# Calibration-free NGS quantitation of mutations below 0.01% VAF

Peng Dai [1,5,6], Lucia Ruojia Wu [1,6], Sherry Xi Chen[1,5], Michael Xiangjiang Wang [1], Lauren Yuxuan Cheng [1], Jinny Xuemeng Zhang[2], Pengying Hao[2], Weijie Yao[2], Jabra Zarka [3], Ghayas C. Issa [3], Lawrence Kwong[4] & David Yu Zhang[1,2 ✉]

Quantitation of rare somatic mutations is essential for basic research and translational clinical applications including minimal residual disease (MRD) detection. Though unique molecular identifier (UMI) has suppressed errors for rare mutation detection, the sequencing depth requirement is high. Here, we present Quantitative Blocker Displacement Amplification (QBDA) which integrates sequence-selective variant enrichment into UMI quantitation for accurate quantitation of mutations below 0.01% VAF at only 23,000X depth. Using a panel of 20 genes recurrently altered in acute myeloid leukemia, we demonstrate quantitation of various mutations including single base substitutions and indels down to 0.001% VAF at a single locus with less than 4 million sequencing reads, allowing sensitive MRD detection in patients during complete remission. In a pan-cancer panel and a melanoma hotspot panel, we detect mutations down to 0.1% VAF using only 1 million reads. QBDA provides a convenient and versatile method for sensitive mutation quantitation using low-depth sequencing.

[1] Department of Bioengineering, Rice University, Houston, TX, USA. [2] NuProbe USA, Houston, TX, USA. [3] Department of Leukemia, The University of Texas MD Anderson Cancer Center, Houston, TX, USA. [4] Department of Translational Molecular Pathology, The University of Texas MD Anderson Cancer Center, Houston, TX, USA. [5] Present address: NuProbe USA, Houston, TX, USA. [6] These authors contributed equally: Peng Dai, Lucia Ruojia Wu.
✉email: genomic.dave@gmail.com

DNA variants with low allelic frequencies have important clinical and biological implications, as they often lead to resistance or recurrence in infection[1,2] and cancer treatments[3–5]. Sensitive genetic testing is highly desired in both minimal residual disease (MRD)[6–8] detection and liquid biopsy[9,10]. Detection of MRD in acute myeloid leukemia (AML) has prognostic and therapeutic implications aimed at preventing morphologic relapse[8]. Sensitive detection of leukemia-specific mutation markers could improve prognostication by identifying submicroscopic disease during remission[6]. Compared to MRD detection by multicolor flow cytometry (MFC)[11,12], NGS MRD assays have the potential for detection of "actionable" mutations to guide therapy selection. The cell-free DNA (cfDNA) in circulation plasma provides a 'snapshot' of dying cells around the body and thus is widely used in liquid biopsy for non-invasive genetic testing. It is frequently the most accessible clinical sample for applications such as therapy selection, post-treatment monitoring, and early cancer screening. Because the tumor-derived DNA is mixed with large amount of normal DNA[13,14], variant allele frequency (VAF) for cancer-related mutations is often low requiring high assay sensitivity.

Polymerase error during amplification[15,16] and sequencing error of NGS platforms[17,18] made it difficult to robustly quantitate low-frequency mutations <1% VAF using conventional NGS technologies. Unique molecular identifiers (UMIs) have been developed to suppress the errors to detect mutations below 0.1% VAF[19,20]. Recent advances in DuplexSeq[21], NanoSeq[22], and SaferSeqS[23] have further reduced errors by grouping both strands of a DNA molecule together into a duplex family to distinguish DNA damage with real mutation achieving confident variant calling at 0.01% VAF or lower. However, since all template molecules, regardless of wild-type (WT) or variant molecules, are sequenced redundantly in current UMI-based methods, they require sequencing to extremely high depths proportional to input molecule amount. On the other hand, high input DNA amount is needed for successful sampling of rare variants. For a mutation with 0.005% VAF, a total of 75,000 diploid human genomic DNA (gDNA) is required to achieve an average of 3.75 mutant copies. This corresponds to approximately 500 ng gDNA. The combination of UMI and high input amount leads to sequencing depth unaffordable for many researchers, clinicians, and patients. Blocker displacement amplification (BDA)[24,25] enriches variant alleles by introducing rationally designed blocker oligonucleotides that competes with forward primer to suppress the amplification of WT molecules. BDA allows detection of rare mutations using low sequencing depth, but loses VAF quantitation without calibration.

To overcome these challenges, herein we have developed QBDA, a method that allows calibration-free accurate VAF quantitation with low-depth sequencing by integrating molecular barcoding with BDA technology for variant enrichment. Because the amplification of WT molecules is suppressed, the number of WT UMI families does not represent actual number of WT molecules. Thus, VAF is calculated based on variant molecule count from QBDA and the input molecule count (i.e., number of input genome copies), which can be calculated from input DNA amount or by adding internal positive control amplicons that quantify a small portion of the input molecules at several different loci in house-keeping genes.

Herein, we demonstrate that mutations within targeted regions are simultaneously enriched and accurately quantified, including single-base substitutions and indels. We apply the QBDA technology to a 20-gene AML panel and demonstrate a robust quantitation of single-base substitutions and indels down to 0.001% VAF at a single locus for MRD analysis. Finally, two QBDA cancer panels including a comprehensive pan-cancer panel and a specific melanoma panel are demonstrated on tumor tissue samples and cfDNA samples.

## Results

**Development of QBDA**. A PCR-based UMI addition approach is performed to attach UMI to each individual DNA single strand in the original DNA templates, followed by BDA to enrich variant amplicons (Fig. 1a). In BDA, a rationally designed blocker DNA oligonucleotide that partially overlaps with the 3′ of the forward primer is introduced to suppress the amplification of WT molecules. The nucleotide sequence unique to the blocker and not in the forward primer is the *enrichment region*; any nucleotide change in this region will prevent the hybridization of blocker to the template, thus allows extension of forward primer.

VAF calculation in QBDA does not require counting WT molecules. In standard UMI-based, non-allele-enrichment NGS methods, the VAF of a mutation call can be calculated as:

$$\mathrm{VAF} = M_\mathrm{v}/M_\mathrm{t} \tag{1}$$

where $M_\mathrm{v}$ is the UMI family count of the mutation, and $M_\mathrm{t}$ is the total number of UMI family count for this locus.

In QBDA, because the amplification of WT is suppressed, the number of WT reads is small and thus UMI count of WT does not represent actual number of WT molecules. Therefore, we calculate $M_\mathrm{t}$ as the following:

$$M_\mathrm{t} = 2 \times w_\mathrm{input} \times c_\mathrm{genome} \times \chi \times N \tag{2}$$

Here $w_\mathrm{input}$ is the amount of input DNA in ng, $c_\mathrm{genome}$ is the number of haploid genomes per 1 ng DNA (for human gDNA, $c_\mathrm{genome} = 300 \, \mathrm{ng^{-1}}$), $\chi$ is the UMI barcoding conversion yield, and $N$ is the copy number of this loci relative to the genome ($N = 1$ for normal loci, >1 for copy number amplification, <1 for copy number loss). We assume $N = 1$ if no CNV data is available. Because two different UMIs are attached to the two strands of one original DNA molecule in QBDA, the number is multiplied by 2.

Based on our observations, the UMI barcoding conversion yield $\chi$ for each amplicon remains consistent across different NGS runs. $\chi$ was characterized using a library prepared from normal DNA ($N = 1$) with QBDA protocol but without the blockers (i.e., no enrichment). From this library, $\chi$ for each amplicon was calculated as:

$$\chi = M_\mathrm{t}/(2 \times w_\mathrm{input} \times c_\mathrm{genome}) \tag{3}$$

The pan-cancer panel further incorporates internal positive control amplicons without blocker into the panel, which quantitates the molecule at several loci in house-keeping genes to estimate the DNA input amount. In pan-cancer panel, $M_\mathrm{t}$ is calculated from the UMI counts of internal positive control amplicons.

**QBDA demonstration**. We first demonstrated the variant enrichment, error correction, and quantitation of QBDA using a single-plex QBDA (Supplementary Table 1 and Supplementary Note 1). Here nine different mutations including single-base substitution, insertion and deletion within an 18 nt region (Supplementary Fig. 1) were enriched using the same BDA primer-blocker set; these mutations are from rpoB (Rv0667) gene of *Mycobacterium tuberculosis*, and are relevant to tuberculosis drug resistance. We mixed H37Rv (WT) DNA with nine synthetic DNA templates each bearing a different mutation to prepare a sample containing ~1% VAF for each of the nine mutations.

QBDA simultaneously enriches mutations and corrected errors. Using standard, PCR-based NGS, the majority of reads (87.6%) were WT, which do not contribute to variant sequencing

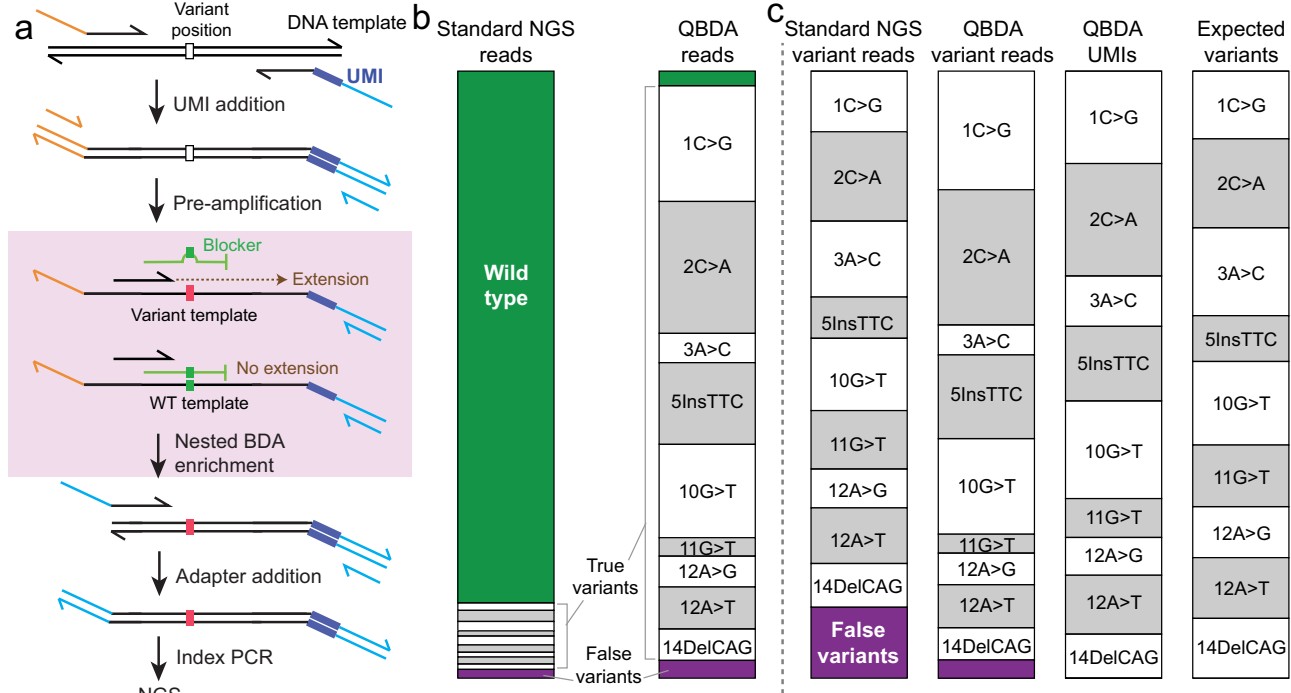

**Fig. 1 Quantitative blocker displacement amplification (QBDA) technology. a** QBDA library preparation workflow. UMIs are attached to DNA templates by two cycles of PCR, followed by pre-amplification using universal primers. Next, a nested BDA was performed to enrich variant sequence. The forward primer is closer to the variant position than the primer in the UMI additions step, in order to suppress primer dimer and nonspecific amplification; an overlapping Blocker suppresses the amplification of wild-type (WT) templates and allows enrichment of variant templates over many PCR cycles. The NGS adapter is added to the enrichment product, followed by index PCR and sequencing. **b** Reducing WT reads by QBDA enrichment. WT DNA was mixed with 9 synthetic DNA gBlocks, each containing a different single-base substitution or indel in a 16 nt region, resulting in about 1% variant allele frequency (VAF) for each mutation. Using standard amplicon-based sequencing without enrichment, 88% reads were used for unnecessary repeated sequencing of WT. Using QBDA, all nine mutations were enriched using a single set of primer and Blocker, and the WT reads are suppressed to 2%. **c** Suppressing error and improving quantitation by UMI in QBDA. Six hundred seventy-five types of false-positive (non-expected) variants were observed in standard NGS in (**b**), occupying 12% of all variant reads, or 1.5% of total reads. All the false variants were removed using UMI-based error correction in QBDA. The observed molecule count for all spike-in variants was within twofold of the expected values.

depth. Using QBDA, sequencing reads became more focused on the mutations, and the WT reads were suppressed to only 2.4% (Fig. 1b). In BDA-based enrichment, the amplification efficiency is not the same for different mutations. Instead of performing calibration curve to obtain the variant enrichment efficiency for all the possible mutations, here we used UMI to improve mutation quantitation accuracy and suppress error (Fig. 1c). In standard NGS, 11.7% of the variant reads did not match the nine expected spike-in mutations, thus were false-positive variants. In QBDA after UMI-based error correction, all the false variants were removed (Fig. 1c, see "Methods" Section for bioinformatics and molecule count calculation). We calculated the counts of unique UMI families for each variant in QBDA, and compared them with expected variant molecule counts. Here the expected variant molecule counts were obtained from a UMI-based NGS library without BDA enrichment. All the observed molecule counts were within twofold of the expected values.

**Multiplexed QBDA quantitation**. We validated QBDA quantitation capability on a 0.1 and 1% VAF sample prepared by mixing repository human cell line DNA sample NA18562 with NA18537. A 10-plex QBDA panel covering ten SNP loci with different genotypes in the two cell line DNA samples was built (Supplementary Fig. 2 and Supplementary Table 2). The calculated VAFs for all the loci were within twofold of expected true value in 1% sample, and seven out of ten were within twofold in 0.1% sample, with the other three were still within threefold (Supplementary

Fig. 2c). Stochasticity in sampling a small number of molecules contributed to quantitation error in 0.1% sample as only 30 ng gDNA is used, corresponding to only nine haploid of variant at 0.1% VAF. Furthermore, variant enrichment does not lead to higher error rate comparing to no enrichment (Supplementary Fig. 2e).

**QBDA AML panel for MRD detection**. To demonstrate quantitation of <0.01% VAF rare mutation for MRD analysis, we next built a 22-plex QBDA panel covering AML-related mutation hotspot regions in 20 different genes for MRD detection (Supplementary Tables 3 and 4). *De novo* mutation calling was performed for all 382 nucleotide positions in 22 enrichment regions; mutations with ≥6 unique UMI families (corresponding to ≥3 original DNA molecules in QBDA) and having VAFs above or equal to the LoD threshold were reported. The LoD threshold is below 0.01% VAF, but varies for different types of mutations (Fig. 2a, Supplementary Note 3.2 for LoD).

Validation of the AML panel was performed using a positive sample containing 22 mutations, which was prepared by mixing PBMC DNA from a healthy donor, Horizon Myeloid DNA Reference Standard, and 3 synthetic DNA templates (Supplementary Note 3.1). The expected VAF was between 0.001% and 0.1%; 16 out of 22 mutations were around 0.01% (between 0.005% and 0.02%). There were 19 single-base substitutions, 2 insertions, and 1 deletion in this positive sample. Using 1 μg of DNA input, all 22 mutations were observed; 82% of the mutations were within

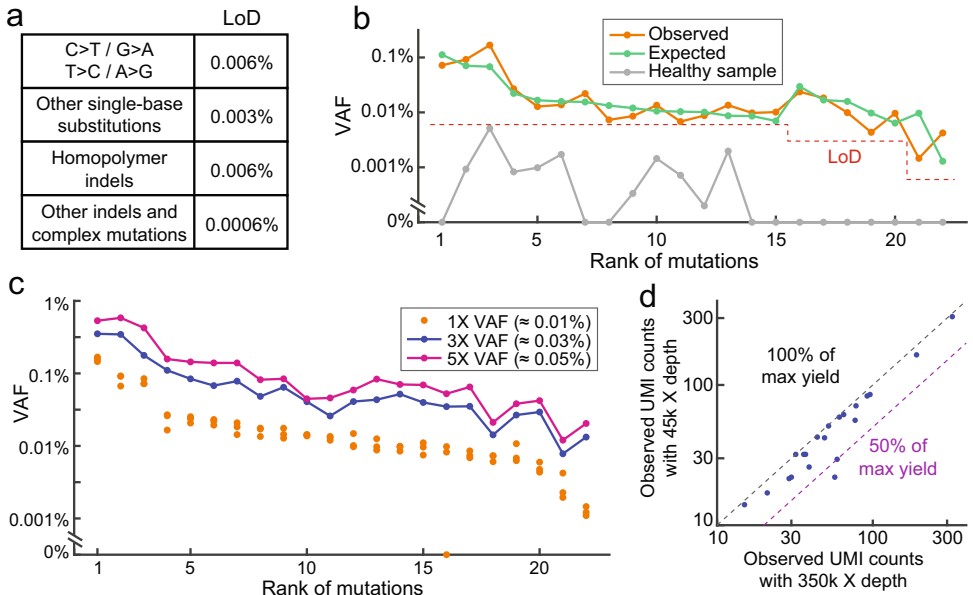

**Fig. 2 Characterization of QBDA AML panel for minimal residual disease (MRD) detection. a** Limit of detection (LoD) threshold for different types of mutations. **b** Observed mutation VAF in a spike-in positive sample and a healthy PBMC sample. The positive sample was prepared by mixing Horizon Myeloid DNA Reference Standard, 3 synthetic gBlocks, and a gDNA sample extracted from healthy PBMC, resulting in VAF between 0.001% and 0.1% for 22 different mutations. Sixteen out of 22 mutations were around 0.01% VAF (between 0.005% and 0.02%). The "expected" VAF was quantitated by UMI-based NGS without mutation enrichment. All 22 mutations covered by the AML panel were observed in the positive sample (orange line); 82% of the mutations were within twofold of expected VAF. The same healthy PBMC sample was also analyzed alone as the paired negative sample using the AML panel (gray line). In a healthy sample, some mutations (C > T or G > A) are observed at below-LoD level, possibly due to clonal hematopoiesis. Here 1 μg of gDNA was used for each library. **c** Quantitation accuracy. The positive sample in (**b**) was sequenced in triplicate NGS libraries; two additional positive samples with threefold or fivefold VAF of the abovementioned sample were also analyzed. For each of the 22 mutations, the observed VAF was in correct order for the 1×, 3×, and 5× VAF samples. In the triplicate experiment of the 1× VAF (≈0.01%) sample, one mutation was not observed in one of the replicates, thus the sensitivity is approximately 1–1/(22 × 3) = 98.5%. One micrograms of gDNA was used for each library. **d** Sequencing depth down to 45,000× does not affect sensitivity in 1X VAF (≈0.01%) sample. The 1× VAF positive sample (500 ng input) was sequenced with 350,000× depth (7.7 M reads). Even after downsampling to 45,000× depth sequencing by random sampling 1.0 M reads from the original library, all mutations are observed. The median observed UMI counts from 20 independent simulations were plotted against observed UMI counts in the original library.

twofold of expected VAF, and 100% were within 1 order of magnitude. Here the expected VAF was quantitated by UMI-based NGS without enrichment. The quantitation is less accurate for some lower VAF mutations, which is likely a result of stochasticity in sampling a small number of DNA molecules (Fig. 2b). The healthy PBMC DNA used in the positive sample was also assayed using the AML panel as a negative control. Using the same input amount (1 μg), none of the 22 mutations was above the LoD threshold in Fig. 2a. In this experiment, the non-zero mutations were all C > T or G > A substitutions, which are possibly results of clonal hematopoiesis[26,27] (Fig. 2b).

Technical sensitivity was analyzed by testing the above-mentioned positive sample in triplicates (1 μg DNA input each). There was only one false negative out of the three libraries, corresponding to 1–1/(22 × 3) = 98.5% technical sensitivity. If we only consider the 16 mutations between 0.005% and 0.02% VAF, the technical sensitivity was 1–1/(16 × 3) = 97.9% (Fig. 2c).

The specificity of AML panel was assessed using a "negative sample". Because QBDA is highly sensitive to mutations below 0.01% VAF, and even healthy blood donors have low-level mutations in their PBMC DNA as a result of DNA damage or clonal hematopoiesis, such as C > T or G > A substitutions[26,27], there is no perfect "negative sample" for MRD detection (Supplementary Fig. 3). We prepared five replicated libraries from the same healthy PBMC gDNA sample to analyze specificity of QBDA AML panel; each library had 1 μg of gDNA input. If a mutation is observed in ≥4 out of the 5 libraries, we believe this is a true positive mutation existing in the DNA sample, not an

artifact caused by polymerase misincorporation or sequencing error, because the probability of the same error appearing 4 times out of 5 experiments is extremely low. After filtering out the true positives, we observed only 1 false-positive mutation call out of the 5 libraries. Therefore, the technical specificity of AML panel can be calculated as 1–1/(382 × 5) = 99.95% at the current LoD threshold, where 382 is the number of enriched nucleotide positions in the panel.

We next prepared samples with threefold or fivefold of the VAF in the abovementioned positive sample. For each of the 22 mutations, higher VAF input always generates higher observed VAF; therefore, we can confidently differentiate samples with 0.02% VAF difference ($p = 3 \times 10^{-6}$ by paired Wilcoxon signed-rank test, Fig. 2c). Sequencing depth down to 45,000× does not affect sensitivity in the 1× VAF (≈0.01%) sample using *in silico* random down-sampling analysis (Fig. 2d). 23,000× depth is still acceptable for detection of 0.01% VAF, but we recommend 45,000× depth for more accurate quantitation (Supplementary Fig. 4).

**Detection of ultralow VAF mutations during AML complete remission.** QBDA AML panel was applied to clinical samples, and was compared with other MRD detection methods including MFC[12] and conventional NGS[28]. Ten paired bone marrow aspirates from five AML patients sampled at diagnosis and during complete remission were tested by QBDA panel. All patients chosen were *NPM1* mutated at diagnosis given that mutations in

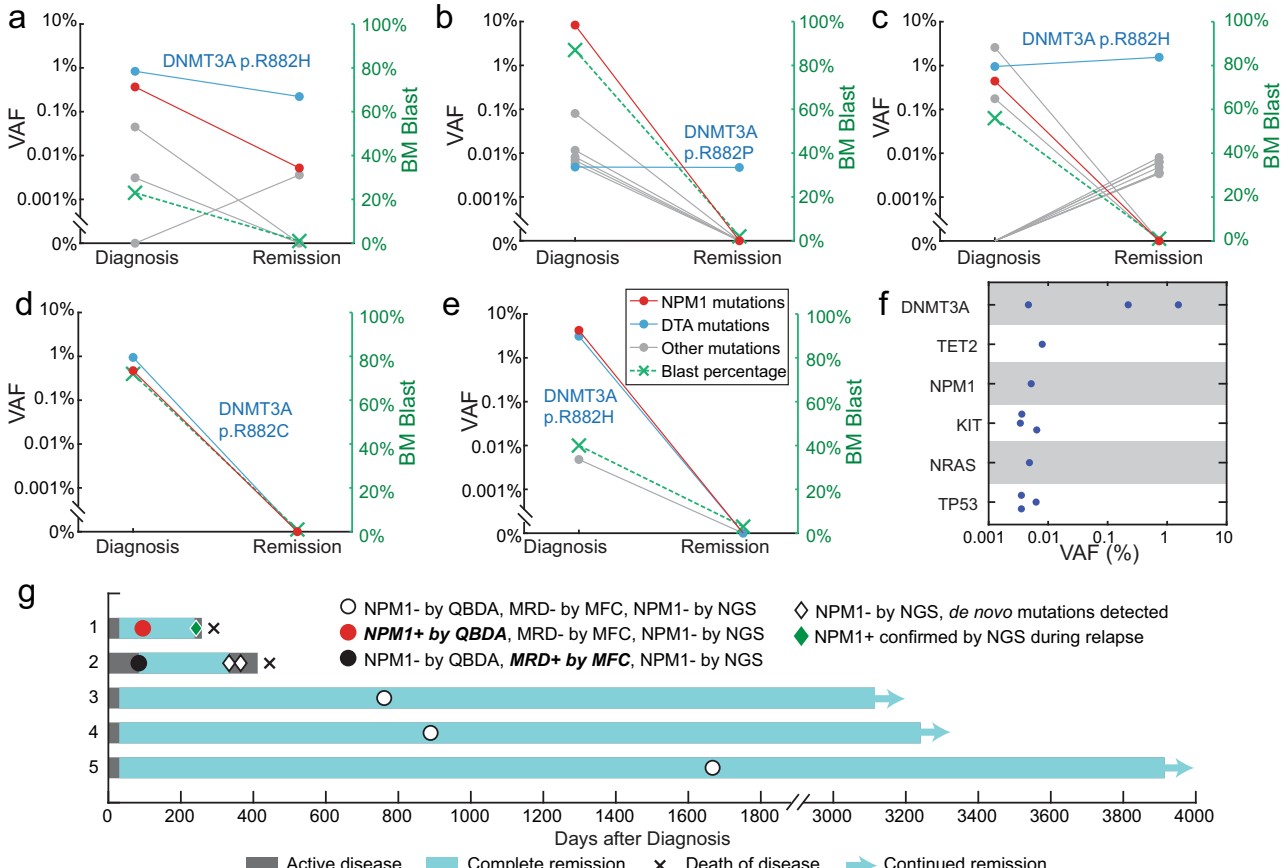

**Fig. 3 QBDA for mutation detection during AML complete remission. a–e** Changes of mutation VAF and the percentage of blasts in bone marrow from diagnosis to complete remission for each of the five patients. The mutations in *NPM1* were highlighted in red and mutations in *DTA* (i.e., *DNMT3A*, *TET2*, and *ASXL1*) were highlighted in blue. Other mutations were shown in gray. **f** Summary of mutations detected from five patients during remission using the QBDA AML panel covering 22 hotspot regions in 20 genes. **g** Swimmer plot of clinical course and molecular findings of patients. QBDA identified *NPM1* mutation in patient 1 during remission while flow cytometry reported MRD negative and conventional NGS failed to detect *NPM1* mutation at the same time point. This *NPM1* mutation was observed by conventional NGS during relapse. QBDA did not observe *NPM1* mutation in patient 2 while MRD positive is reported by flow cytometry. In the two subsequent time points even after relapse *NMP1* mutation was still not observed. Instead, *de novo* mutations in *KDM6A* and *PHF6* were identified indicating clonal evolution occurred as alternative cause of relapse.

*NPM1* are considered founder mutations in the pathogenesis of AML[29] and *NPM1* is a validated MRD marker[6].

Mutation VAF and the percentage of blasts in bone marrow at diagnosis and during remission for each of the five patients were plotted (Fig. 3a–e). The allele frequencies for mutations detected in five patients during remission were summarized (Fig. 3f). A full list of mutations and patient information were summarized in Supplementary Tables 5 and 6. Persistent mutations were detected in three out of the five patients. Preleukemic mutations in the epigenetic regulators *DTA* (i.e., *DNMT3A*, *TET2*, and *ASXL1*) were most common and were observed in all three patients with mutations detected during remission. This is consistent with previous observations that they are often present in persons with age-related clonal hematopoiesis, and are not significantly associated with increased relapse risk[7,30–35]. Other mutations observed during remission include *NPM1*, *KIT*, *NRAS*, and *TP53*.

A swimmer plot of clinical course and molecular findings of each patient is summarized (Fig. 3g). QBDA identified NPM1 mutation in only one patient (patient #1) during remission at a VAF of 0.0052%. In spite of the low allele frequency detected, the duration of remission is only 7.0 months for this patient. However, flow cytometry reported MRD negative and conventional NGS failed to detect *NPM1* mutation at the same time point for this patient. This NPM1 mutation was confirmed by

conventional NGS at relapse, indicating QBDA's accuracy of rare mutation detection and potential of early detection.

QBDA reported no *NPM1* mutation during remission in the other four patients which is in concordance with conventional NGS. Three of them were MRD negative by flow cytometry, with over 100 months of remission (patient #3~5). In one case, however, MRD positive is reported by flow cytometry and the duration of remission is 8.1 months (patient #2). *NPM1* mutation was not observed in the two subsequent time points even after relapse using conventional NGS. Instead, de novo mutations in *KDM6A* and *PHF6* were identified. We thus believe that QBDA is accurate in reporting no *NPM1* mutation during remission but clonal evolution occurred as alternative cause of relapse[29,36]. QBDA assay allows sensitive detection of rare mutations in genes of interest, which we envision to be significant for relapse risk assessment.

**QBDA pan-cancer panel for MRD detection.** Next, we demonstrated highly multiplexed QBDA to simultaneously detect variants in 180 amplicons per tube. VarMap™ Pan-Cancer NGS Panel from NuProbe Inc. was developed based on QBDA technology, which covers 61 genes and 360 hotspot regions in two tubes (Supplementary Fig. 5). It is compatible with MRD

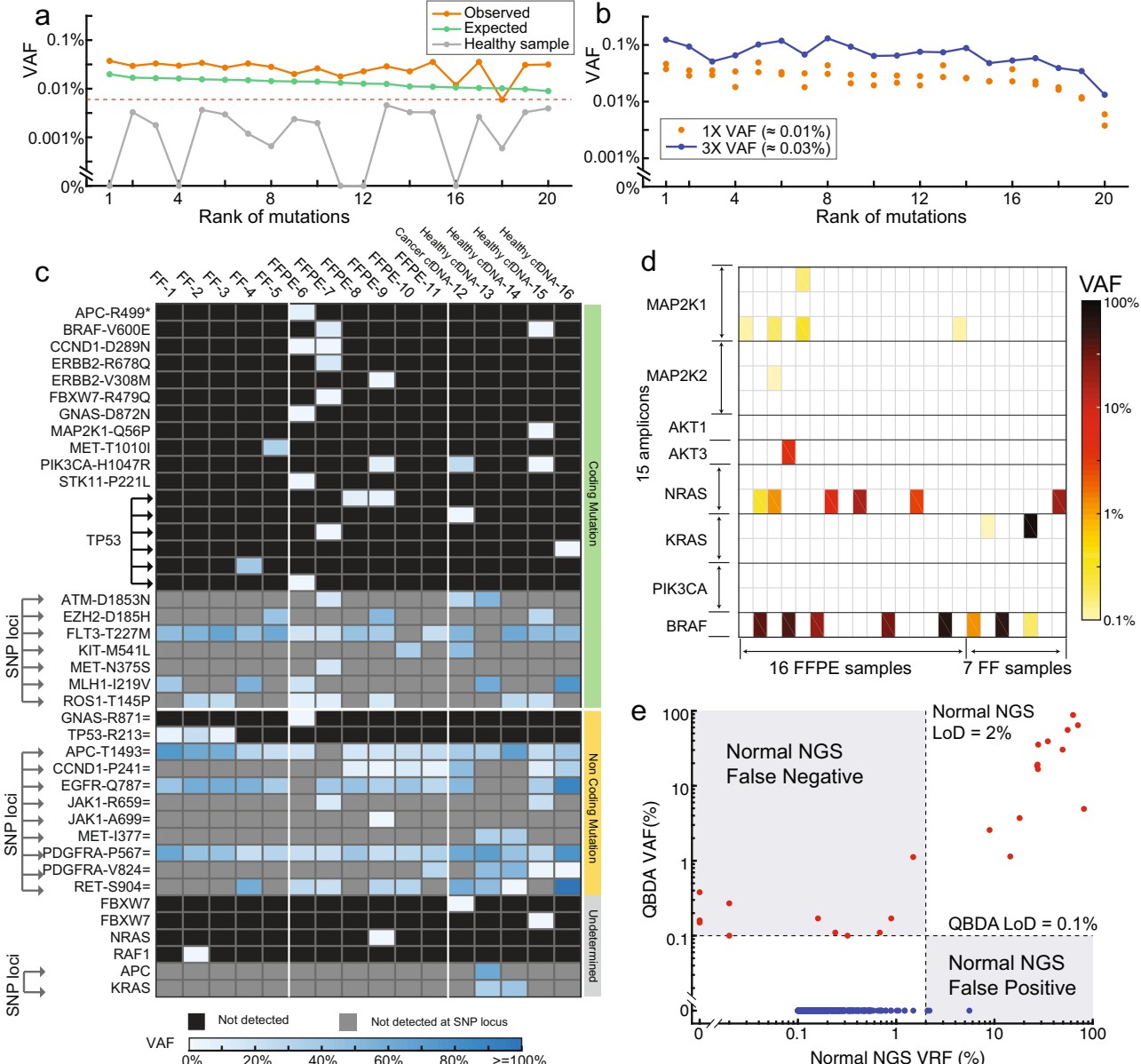

**Fig. 4 Application of QBDA technology to pan-cancer hotspot large panel and Melanoma panel. a** Compatibility of pan-cancer panel with ultralow frequency mutation analysis. Observed mutation VAF in a spike-in positive sample (VAF ≈ 0.01%) and the negative sample without spike-in are plotted. With mutation calling threshold setting at 0.006% VAF, the technical sensitivity was 95% based on duplicate test of spike-in positive sample. **b** Quantitation accuracy. The positive sample in (**a**) was sequenced in duplicate NGS libraries; one additional positive sample with threefold VAF of the abovementioned sample was also analyzed. For each of the 20 spike-in mutations, the observed VAF was in correct order for the 1× and 3× samples. **c** Pan-cancer panel for quantitation of mutations down to 0.1% VAF using 1 M reads (mean sequencing depth 2800X) in 16 clinical samples including FFPE, Fresh Frozen (FF) and cfDNA. Three hundred sixty amplicons in hotspot regions of 61 genes are tested and only detected mutations are plotted here. **d** Melanoma panel for detection of mutations down to 0.1% in clinical samples. VAF of observed mutations in 23 FFPE or FF clinical samples from Melanoma patients are summarized. Co-existence of *BRAF* V600E and low-frequency *NRAS* Q61K mutations in FFPE5 sample was observed. **e** QBDA quantitation exhibits less false negative and false-positive variant calls than normal NGS without UMI. All observed variants in the 23 melanoma clinical samples are plotted.

detection at 0.01% VAF using 1 µg DNA input and 25 M reads per tube. Validation was performed similarly as AML panel using a positive sample containing 20 mutations, which was prepared by mixing PBMC DNA from a healthy donor and 20 synthetic DNA templates (Supplementary Table 7). All 20 mutations were observed; 60% of the mutations were within twofold of expected VAF, and 100% were within one order of magnitude (Fig. 4a). One of the spike-in mutation is observed in the healthy DNA in all five technical replicates at about 0.016% VAF, which we

consider a true positive mutation existing in the healthy DNA sample. This background is subtracted from the reported VAF in positive samples. Technical sensitivity was analyzed by testing the abovementioned positive sample in duplicates. Setting LoD threshold at 0.006% VAF, there were two false negative out of the two libraries, corresponding to $1-2/(20 \times 2) = 95\%$ technical sensitivity. The healthy PBMC DNA used in the positive sample was also assayed as a negative control. Calculated similarly as AML panel, the specificity of pan-cancer panel is 99.997% with

only 1 false positive mutation with >0.006% VAF detected in five replicate libraries. We next prepared samples with threefold of the VAF in the abovementioned positive sample. For each of the 20 mutations, higher VAF input always generate higher observed VAF (Fig. 4b).

**Low-depth sequencing with pan-cancer panel and melanoma panel.** In liquid biopsy samples, the available DNA amount is in ng range and thus too low for detecting 0.01% VAF mutations. In tumor tissue samples, the background mutation derived from formalin-fixed paraffin-embedded (FFPE) DNA damage is often higher than 0.1% VAF. Therefore, for tissue DNA or liquid biopsy analysis, an assay with an LoD of 0.1% VAF using low input DNA and low sequencing depth is more desired than an extremely sensitive assay requiring high input and high sequencing depth. QBDA allows detection of mutations down to 0.1% VAF from FFPE and blood plasma specimens with as low as 6–20 ng input, which can potentially help to understand resistance mechanism and clonal evolution to guide treatment.

We first applied a comprehensive QBDA-based cancer mutation panel, the VarMap™ Pan-Cancer Panel, for quantitating mutations above 0.1% VAF with 10 ng DNA input and 0.5 M reads per tube. We tested 16 samples, including 6 FFPE DNA samples from breast, colorectal or lung cancer patients, 5 fresh frozen (FF) DNA from hepatocellular carcinoma patients, 1 plasma cfDNA from breast cancer patients and 4 cfDNA from healthy people (Fig. 4c and Supplementary Table 8). On average, 1.9 somatic mutations at non-SNP loci were detected per sample. Because QBDA allows low-depth detection of low-frequency mutations, all the 16 samples can be sequenced in one Miniseq or Miseq run, enabling tissue or liquid biopsy pan-cancer genetic tests with de-centralized sequencing instruments.

Next, a QBDA melanoma panel (Supplementary Note 5) was applied to 16 FFPE and 7 FF clinical tissue samples (Fig. 4d, Supplementary Table 11), and we found co-existence of *BRAF* V600E and low-frequency *NRAS* Q61K mutations in one FFPE tissue. Although *BRAF* and *NRAS* mutations are usually mutually exclusive in melanoma patients, *BRAF/NRAS* dual mutation may derive from two subclonal populations. As the patient was treated with *BRAF* inhibitor, co-existence of low frequency *NRAS* indicated potential clonal evolution and resistance mechanism related to *NRAS*. Consistent with our observation, there were recent reports in which *BRAF* and *NRAS* co-mutations were observed in the same cell after treated with a *BRAF* inhibitor[37].

## Discussion

Considering the molecular heterogeneity of AML, MRD analysis based on mutation biomarkers in bone marrow DNA could provide *actionability* to guide treatment decision as a complementary method for MFC and morphology-based assessment of remission. The sensitivity and cost for NGS MRD analysis are dependent on the assay's analytical LoD and sequencing depth, respectively. QBDA combines variant enrichment with molecular barcoding in NGS to allow detection of mutation down to 0.001% VAF with about 23,000× sequencing depth. When applied to clinical samples, QBDA identified residual *NPM1* mutation at 0.005% VAF in one patient during remission while both flow cytometry and conventional NGS failed to detect MRD at the same time point for this patient. The accuracy of QBDA mutation call was supported by clinical outcome of short duration of remission as well as confirmation of such mutation at relapse by conventional NGS, indicating QBDA's potential of early detection.

QBDA quantitation is accurate. We extensively validated quantitation accuracy by comparing QBDA VAF with spike-in ratio of cell line DNA or synthetic template, with expected allele frequencies in commercial myeloid DNA Reference Standard, with VAF from digital droplet PCR (ddPCR), and with conventional NGS. QBDA reduced both false-positive and false-negative variant calls comparing to conventional NGS in the 23 melanoma clinical samples (Fig. 4e). Validation against ddPCR was performed in clinical DNA samples with BRAF/NRAS mutations (Supplementary Fig. 7, Supplementary Table 12). To validate no false-negative call were made, one healthy donor PBMC gDNA sample and three FFPE samples without *BRAF/NRAS* mutation by QBDA were also tested by ddPCR and were confirmed with no mutation (Supplementary Fig. 8, Supplementary Table 12). Remaining errors in quantitation may be due to Poisson distribution in sampling or DNA damage. We introduced different UMI sequences to each strand of DNA molecule by PCR and thus duplex family information is lost during denaturation of UMI attachment PCR. We expect that error from DNA damage may be further suppressed in QBDA by using ligation-based UMI attachment, so that in downstream bioinformatics analysis both strands of a DNA molecule can be grouped into a duplex family similar to DuplexSeq[21], NanoSeq[22], and SaferSeqS[23] while still reducing sequencing depth by BDA variant enrichment.

The gene ploidy impacts VAF in QBDA, but QBDA is able to accurately quantitate VAF in case CNV and mutation are simultaneously present in the gene of interest as long as copy number for the gene is normalized. As demonstrated in the formula of calculating total number of UMI family count for each locus ($M_t$), $M_t$ needs to be adjusted by the copy number in genome if CNV occurs. As an example of copy number normalization, *BRAF* gene in melanoma FFPE12 sample underwent both copy number variation (CNV) and mutation; VAF for *BRAF* V600K mutation was consistent with ddPCR after normalizing the copy number of *BRAF* gene (Supplementary Fig. 7). The copy number of gene of interest can be measured by ddPCR, whole-genome or whole-exome sequencing. We further demonstrate QBDA panel without blocker can be used for CNV calculation (Supplementary Fig. 9) and the ploidy of BRAF in FFPE12 is highly consistent with ddPCR result.

Broad coverage, mutation sensitivity, and low sequencing cost are simultaneously explored by the 61-gene pan-cancer QBDA panel that detects mutations down to 0.1% VAF requiring only 1 M reads per sample, or detects MRD at 0.01% VAF using 1 µg DNA input and 50 M reads per sample. We envision MRD based on large Pan-Cancer panel can pick up *de novo* drug resistance mutations to guide treatment decisions based on its high coverage.

## Methods

**QBDA protocol**. QBDA Library preparation consisted of three PCR reactions (Fig. 1a): UMI addition and pre-amplification, BDA for variant enrichment, and index PCR, all performed on a T100 Thermal Cycler (Bio-Rad). First, DNA sample was mixed with the specific forward primer (SfP), specific reverse primer (SrP) and amplified using high fidelity Phusion polymerase. The final concentration for each SfP and SrP was 15 nM unless otherwise noted. Two cycles of long-extension PCR were performed for the addition of UMI on all target loci, followed by a universal amplification. In order to amplify the molecules to avoid sample loss during purification while preventing addition of multiple UMIs onto the same original molecule, the annealing temperature was raised with short annealing time (30 s) with universal forward primer (UfP) and universal reverse primer (UrP). The addition of UfP and UrP into the reaction was an open-tube step on the thermocycler to prevent temperature drop and primer dimer formation. Thermal cycling condition was: 98 °C:30 s − (98 °C:10 s − 63 °C:30 min − 72 °C:60 s) × 2 − (98 °C:10 s − 63 °C:20 s − 72 °C:60 s) × 2 − (98 °C:10 s − 71 °C:20 s − 72 °C:60 s) × 5 − (72 °C:5 min) − 4 C:hold. During the last 5 min of the second 30 min at 63 °C, 1.5 µM of each universal primer was added while keeping the reactions inside the thermal cycler. If the DNA input is less than 500 ng, the reaction mixture was purified using AMPure XP beads (1.6× ratio) twice to remove single-stranded primers. If the DNA input is over 500 ng, double-side size selection (0.3×, 1.6× ratio) was performed to remove long input gDNA, followed by another 1.6× AMPure XP beads purification.

Second, BDA amplification was performed. BDA forward primer, BDA blocker, Phusion polymerase, dNTPs, and PCR buffer were mixed with the purified PCR product for BDA amplification. Thermal cycling condition was: 98 °C:30 s − (98 °C:10 s − 63 °C:5 min − 72 °C:60 s) × 23 − 4 °C:hold. The reaction mixture was purified using AMPure XP beads (1.8× ratio).

Next, Adapter is added. BDA adapter primer (Adp_fP, comprising illumina adapter sequence and BDA forward primer sequence) and UrP are mixed with the purified PCR reaction mixture and amplified. Thermal cycling condition was: 98 °C:30 s − (98 °C:10 s − 63 °C:5 min − 72 °C:1 min) × 2 − 4 C:hold. The reaction mixture was purified using AMPure XP beads (1.6× ratio). Lastly, standard NGS index PCR is performed. Libraries are normalized and loaded onto an Illumina sequencer.

**Samples**. FF tissue samples were purchased from OriGene Technologies, Inc. in de-identified format. Sixteen FFPE samples of patients with metastatic stage IV melanoma and ten bone marrow aspirates samples of patients with AML in de-identified format were obtained from MD Anderson Cancer Center. All procedures performed in studies involving human participants were approved by Institutional Review Board at MD Anderson, and were in accordance with the 1964 Helsinki declaration and its later amendments or comparable ethical standards. Informed consent was obtained from all participants. FFPE samples from breast, colorectal and lung cancer patients were purchased from OriGene Technologies, Inc. in de-identified format. Plasma from healthy people were purchased from Zen-Bio Inc. Plasma from breast cancer patients were purchased from Discovery Life Science.

NA18537 and NA18562 DNA were purchased from Coriell Institute for Medical Research. Myeloid DNA Reference Standard was purchased from Horizon Discovery. DNA input was quantified by qubit for gDNA, by qPCR for fragmented DNA (FFPE DNA and cfDNA) to identify the amplifiable portion. Oligonucleotides and synthetic DNA templates (gBlock) were ordered from Integrated DNA Technologies.

**NGS data preprocessing**. The QBDA libraries were analyzed using 130 nt + 21 nt paired-end sequencing on Illumina sequencers. Adapter sequences were removed from read 1 (130 nt), and UMI sequences were extracted from read 2 (21 nt). The processed read 1 sequences were then aligned to designed BDA amplicons using the Bowtie2 software[38].

**UMI-based mutation calling**. Next, reads aligned to each BDA amplicon were grouped by UMI. Reads carrying the same UMI sequence are amplified presumably from the same original DNA template, thus belong to the same UMI family. If the UMI sequence contained unexpected bases that do not match the expected format ($H_{15}$), the UMI family was removed.

Because small UMI family size (i.e., number of reads in the UMI family) might be a result of amplification or sequencing error in the UMI region, UMI families with small family size are removed. To adjust for the difference derived from sequencing depth, we use a "dynamic cutoff" to remove small UMI families. If the family size was ≤3 or smaller than 5% of the mean of top 3 family size in the same amplicon, the UMI family was removed.

We next performed *do novo* variant call for each BDA enrichment region. In an effective NGS read, the forward primer and the 10 nt after the enrichment region need to match the corresponding regions in the BDA amplicon. The consensus sequence of each UMI family was the enrichment region sequence appearing most often in the UMI family. If two sequences had the same frequency and were the most common, consensus seuence was arbitrarily selected from these two. The consensus sequences were then compared to the WT enrichment region, and variants were recorded.

**Mutation filtering by UMI count**. Polymerase error may occur during the PCR cycle of UMI attachment. In order to minimize false positives, we applied UMI count filter and VAF filter to remove mutation calls that are less likely clinically relevant. The UMI count filter removes mutation calls with <6 UMI family count; and the VAF filter removes mutation calls with lower than defined LoD threshold. The count filter and VAF filter aim to address potential polymerase mis-incorporation errors, sequencing errors, potential DNA damage, and clonal hematopoiesis.

**Digital droplet PCR**. Digital PCR was performed using Bio-Rad QX200 Droplet Digital PCR System. Mutation VAF was confirmed using BioRad ddPCR NRAS Q61K Kit (BioRad Assay ID: dHsaMDV2010067) and BioRad ddPCR BRAF V600 Screening Kit (Catalog # 12001037). Copy number of BRAF was confirmed with BRAF CNV FAM assay (BioRad Assay ID: dHsaCP2500366) and EIF2C1 (Ref) HEX assay (BioRad Assay ID: dHsaCP2500349). Data were analyzed using Bio-Rad Quantasoft Software v1.4.

**Conventional NGS for AML clinical samples**. DNA was extracted from bone marrow samples and NGS was performed on clinical-grade, Clinical Laboratory Improvement Amendments-compliant platforms using an Illumina MiSeq system (Illumina, Inc., San Diego, CA, USA). The NGS panels included genes frequently affected in hematologic malignancies (panels of 28, 53, or 81 genes developed at MD Anderson[28]; see Supplementary Data 8 for the full list of genes). A minimum sequencing coverage of ×250 (bidirectional true paired-end sequencing) was required. The analytical sensitivity was established at 5% mutant reads on a background of WT reads.

**Reporting summary**. Further information on research design is available in the Nature Research Reporting Summary linked to this article.

## Data availability
The sequences of the DNA oligonucleotides used for QBDA panels, QBDA test results and relevant de-identified clinical sample information are included in Supplementary Information and Supplementary Data 1–8. Raw sequencing data for QBDA AML panel has been deposited at NCBI BioProject ID PRJNA767049, and can be found at https://doi.org/10.6084/m9.figshare.15117642.v1 and https://doi.org/10.6084/m9.figshare.15102276.v1. Source data are provided with this paper.

## Code availability
NGS data analysis pipeline for QBDA variant calling is available from Github (https://github.com/wrj915/QBDA).

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

## Acknowledgements

This work was supported by NIH awards U01CA233364 and R01CA203964 to D.Y.Z., and CPRIT award RP180147 to D.Y.Z. This work is also supported by Nuprobe USA.

## Author contributions

P.D., L.R.W. and D.Y.Z. conceived the project. P.D. and L.R.W. designed and conducted the experiments, and analyzed the data. S.X.C. and M.X.W. analyzed the data. L.Y.C. designed a melanoma panel. J.X.Z. and P.H. performed pan-cancer panel experiments. W.Y. performed NGS experiments. J.Z. and G.C.I. provided clinical AML samples and analyzed the data. L.K. provided melanoma clinical samples and analyzed the data. P.D. L.R.W. and D.Y.Z. wrote the paper with input from all authors.

## Competing interests

A US provisional patent application (No. 63/018,922) and an international patent application (No. PCT/US2021/030249) covering the use of QBDA technology have been filed in which the Rice University is the applicant, and D.Y.Z., L.R.W., P.D. and S.X.C. are the inventors. P.D., L.R.W., S.X.C., M.X.W. and L.Y.C. declares a competing interest in the form of consulting for Nuprobe USA. D.Y.Z. declares a competing interest in the form of consulting for and significant equity ownership in Nuprobe USA, Torus Bio-systems, and Pana Bio. The remaining authors declare no competing interests.

## Additional information



