## [Peer Review File · Nature Communications]

Reviewers' Comments:

Reviewer #1:

Remarks to the Author:

- Please expand on strategy to address unique copy number alterations at each locus to be interrogated ("N") particularly in a panel encountering a diverse range of cancer specimens with unique CNVs
- The incorporation of more validation samples, especially FFPE specimens of different quality (age, decal, etc) will help to highlight the potential of this technology
- A section devoted to the more practical aspects of QBDA in FFPE specimens would be helpful. For example, the impact of resolving low allelic events from low input material (10ng).
- Line 339 – Typo – "illumine" should be "Illumina"

Stephen Yip

Reviewer #2:

Remarks to the Author:

Dai and Wu et al. present a method incorporating UMI barcoding and blocker displacement amplification to more accurately determine variant allele frequency. This is applied to the detection of minimal residual disease in AML and other cancers. If adapted the approach would allow for more accurate and inexpensive detection of these low frequency variants. I have two major points that I would like to see addressed:

1) The authors present a method for estimating the total UMI family count for a locus (Mt). This is essential to determine an accurate VAF. I do not believe that the authors properly perform this normalization. First, estimation of $w_{input} * c_{genome}$ is critical to the accurate estimation of total amplifiable DNA - the authors account for this properly only in the pan-cancer panel where they can directly estimate these counts based on internal positive controls. The estimated count is likely to be very sensitive to the DNA prep which may be highly variable across individuals performing the experiment. Second, I am unclear what X as barcoding conversion yield is meant to represent. Based on how this is calculated however, it should represent any bias derived from the efficiency of polymerase blocking, and GC-content bias in the PCR/NGS, and influence of the variant type on these factors. This makes me a bit skeptical of the authors claim (for which they should provide data) that X is consistent across NGS runs. I would anticipate this normalization value to be site-specific as type of variant (homopolymer indel, SNV, multi-base insertion, etc) would likely affect the binding of the blocker and I would absolutely expect the GC content of the region to drive amplification biases at individual regions.

2) The authors also present a nice control of healthy samples in Figures 2 and 4. However, I am a bit perplexed by the authors claim that in a healthy PBMC they find 10/22 known AML-associated variants (and 15/20 of the pan-cancer panel) and say that this is possible due to clonal hematopoiesis. While these mutations fall below the authors self-defined level of detection threshold, I think that this represents some sort of false-positive artifact that needs to be addressed further. How are the exact mutations that are in the cancer panels all appearing in this healthy individual?

Reviewer #1 (Remarks to the Author):

- Please expand on strategy to address unique copy number alterations at each locus to be interrogated (“N”) particularly in a panel encountering a diverse range of cancer specimens with unique CNVs

We thank the reviewer for the comment and suggestion. The DNA from a diverse range of cancer specimens can be split for QBDA and CNV quantitation. The copy number of gene of interest can be measured using any existing method including ddPCR, WES or WGS. Here we performed additional experiments to demonstrate a strategy of addressing CNV using QBDA panel without blocker. Without variant enrichment, the molecule count difference at each locus is due to copy number change. We showed the BRAF copy number in FFPE12 is identified accurately.

Target 15 is an amplicon in BRAF so that BRAF ploidy can be calculated as the ratio between molecule count of target 15 and the mean molecule count of target 1-14. The ratio of BRAF/Ref (2.48) calculated from QBDA in FFPE 12 is consistent with ddPCR result (2.39). In a normal genomic DNA (NA18537) the ratio of BRAF/ref is close to 1.

In discussion section, we added: “The copy number of gene of interest can be measured by ddPCR, whole genome or whole exome sequencing. We further demonstrate QBDA panel without blocker can be used for CNV calculation (Supplementary Fig. S9) and the ploidy of BRAF in FFPE 12 is highly consistent with ddPCR result.”

In SI, we added Figure S9:

Figure S9. BRAF ploidy calculation using QBDA Melanoma panel without blocker. Target 15 is an amplicon in BRAF. The BRAF ploidy is calculated as the ratio between molecule count of target 15 and the mean molecule count of target 1-14. The ratio of BRAF/Ref (2.48) calculated from QBDA in FFPE 12 is consistent with ddPCR result (2.39). In a normal genomic DNA (NA18537) the ratio of BRAF/ref is close to 1.

- The incorporation of more validation samples, especially FFPE specimens of different quality (age, decal, etc) will help to highlight the potential of this technology

We thank the reviewer’s comments. To validate the utility and potential of QBDA technology, we tested a total of 22 FFPE samples from breast, colorectal, lung cancer patients or melanoma patients, and also tested a variety of other clinical samples including bone marrow DNA, cell-free DNA, PBMC DNA and fresh/frozen tissue DNA.

We would like to pursue deeper on the potential of QBDA technology in collaboration with clinicians in the next papers, and really appreciate the suggestion of including more FFPE specimens of different quality (age and decalcification).

- A section devoted to the more practical aspects of QBDA in FFPE specimens would be helpful. For example, the impact of resolving low allelic events from low input material (10ng).

We thank the reviewer's comments. QBDA allows detection of mutations down to 0.1% variant allele frequency from FFPE specimens with as low as 6-20 ng input, which can potentially help to understand resistance mechanism and clonal evolution to guide treatment. As an exemplary case study (Melanoma FFPE5), uncommon co-existence of *BRAF* V600E and low frequency *NRAS* Q61K mutations was observed by QBDA.

Although *BRAF* and *NRAS* mutations are usually mutually exclusive in melanoma patients, *BRAF/NRAS* dual mutation may derive from two subclonal populations. As the patient was treated with *BRAF* inhibitor, co-existence of low frequency *NRAS* indicated potential clonal evolution and resistance mechanism related to *NRAS*. Consistent with our observation, there were recent reports in which *BRAF* and *NRAS* co-mutations were observed in the same cell after treated with a *BRAF* inhibitor (doi: 10.18632/oncotarget.12848.)

We added more details and discussion on the FFPE5 case study and QBDA's potential benefit of identifying drug resistance mutations to guide treatment decisions in section "Low-depth sequencing with pan-cancer panel and melanoma panel", Discussion section in manuscript, and Supplementary Section 5 in SI.

- Line 339 – Typo – "illumine" should be "Illumina"
- We thank the reviewer's comments and have corrected the typo.

Stephen Yip

Reviewer #2 (Remarks to the Author):

Dai and Wu et al. present a method incorporating UMI barcoding and blocker displacement amplification to more accurately determine variant allele frequency. This is applied to the detection of minimal residual disease in AML and other cancers. If adapted the approach would allow for more accurate and inexpensive detection of these low frequency variants. I have two major points that I would like to see addressed:

1) The authors present a method for estimating the total UMI family count for a locus (M_t). This is essential to determine an accurate VAF. I do not believe that the authors properly perform this normalization. First, estimation of $w_{input} * c_{genome}$ is critical to the accurate estimation of total amplifiable DNA - the authors account for this properly only in the pan-cancer panel where they can directly estimate these counts based on internal positive controls. The estimated count is likely to be very sensitive to the DNA prep which may be highly variable across individuals performing the experiment.

Second, I am unclear what X as barcoding conversion yield is meant to represent. Based on how this is calculated however, it should represent any bias derived from the efficiency of polymerase blocking, and GC-content bias in the PCR/NGS, and influence of the variant type on these factors. This makes me a bit skeptical of the authors claim (for which they should provide data) that X is consistent across NGS runs. I would anticipate this normalization value to be site-specific as type of variant (homopolymer indel, SNV, multi-base insertion, etc) would likely affect the binding of the blocker and I would absolutely expect the GC content of the region to drive amplification biases at individual regions.

We thank the reviewer for the comments and suggestions. We compared w_{input} (the amount of input DNA in ng) measure by Qubit or qPCR with the input amount (in ng) calculated based on internal positive control (IPC) amplicons in pan-cancer panel for all the 16 DNA samples reported. As summarized in Table 1 below, the inferred input amounts based on IPC in both tubes of pan-cancer panel are mostly consistent with Qubit or qPCR.

Table 1. Pan-cancer panel DNA input quantitation by different methods

Sample ID	Library input (ng)	Quantitation Method	IPC report (ng), Tube 1	IPC report (ng), Tube 2
FF4146	20.0	Qubit	24.7	27.8
FF4934	20.0		26.4	34.4
FF3176	20.0		27.6	31.0
FF4850	20.0		31.2	36.1
FF4927	20.0		32.6	38.7
FFPE25	10.5	qPCR	25.4	28.1
FFPE26	11.2		25.8	27.2
FFPE23	20.0		28.5	32.0
FFPE20	20.0		28.0	31.3
FFPE6	20.0		26.1	27.9
FFPE5	20.0		29.8	34.3
DLS4 (cfDNA)	10.0		9.4	10.7
cfDNA sample A	10.0	Qubit	8.2	10.3
cfDNA sample C	10.0		9.6	11.9
cfDNA sample D	10.0		8.1	7.8
cfDNA sample E	10.0		13.1	13.6

Conversion yield χ reflects the barcoding yield, i.e. the percentage of original molecules that can be attached with a UMI. The PCR efficiency in the first two cycles of PCR (UMI incorporation step) determines the value of χ . Bias after the UMI incorporation step will only impact reads distribution in the final library and thus influence the UMI family size (the number of reads bearing the same UMI); after UMI-based data processing, bias introduced in the later PCR steps will be corrected, and χ values will not be impacted.

Polymerase blocking by blocker does not impact conversion yield χ as well because it is after UMI attachment step. Blocker will influence the PCR yield per cycle for WT and variant, and for different types of variant molecules during BDA step and thus influence the final reads distribution in the sequencing library. We added Figure S10 in the SI to demonstrate this.

Figure S10. Influence on conversion yield and amplification bias of each step in QBDA.

Intra-operator (Figure S11) and inter-operator (Figure S12) experiments were added in SI to demonstrate the consistency of conversion yield χ between runs. High reproducibility was observed in triplicates of the same operator. Good inter-operator reproducibility is observed as well without optimization of operation difference, especially considering one operator has never performed QBDA previously. We think the inter-operator difference may derive from small difference in setting up the first PCR reaction from person to person and may be further reduced.

Figure S11. Intra-operator conversion yield reproducibility of QBDA Melanoma panel in triplicate experiments.

Figure S12. QBDA Melanoma panel conversion yield is generally consistent between two different operators.

Conversion yield χ is a parameter for amplicon so that it is amplicon-specific; it should not be sensitive to the type of variants that influences blocker binding. Accurate quantitation of different types of mutations in the same amplicon (and same enrichment region) is observed in both TB panel and SNP panel, where different SNPs and indels up to 3-bp are tested. Although it is possible that variant especially long indel might impact χ due to significant change in sequence length and properties, the observation in QBDA quantitation accuracy indicates this would not introduce a quantitation error over 2-fold (Fig. 1c).

2) The authors also present a nice control of healthy samples in Figures 2 and 4. However, I am a bit perplexed by the authors claim that in a healthy PBMC they find 10/22 known AML-associated variants (and 15/20 of the pan-cancer panel) and say that this is possible due to clonal hematopoiesis. While these mutations fall below the authors self-defined level of detection threshold, I think that this represents some sort of false-positive artifact that needs to be addressed further. How are the exact mutations that are in the cancer panels all appearing in this healthy individual?

We thank the reviewer for the comments. We think the observed mutation with VAF lower than LoD in healthy PBMC DNA in the AML study are most likely due to clonal hematopoiesis or DNA damage for two reasons:

(1) Although 10 mutations were reported out of the 22 targets, all the 10 non-zero mutations (Fig. 2b) were all C>T or G>A substitutions, which are the dominant mutation types in clonal hematopoiesis. In a recent study (Clonal haematopoiesis harbouring AML-associated mutations is ubiquitous in healthy adults, doi: 10.1038/ncomms12484) Young *et al.* observed clonal hematopoiesis, frequently harboring mutations in DNMT3A and TET2, in 95% of individuals studied using a sequencing method with 0.03% VAF LoD. C>T or G>A substitutions are the most dominant observed mutation in clonal haematopoiesis in this study. Because LoD for QBDA Leukemia panel is further improved by 10- to 100-fold comparing to previous study, it is likely the known AML-associated mutations in “healthy” PBMC DNA are existing in the sample as opposed to random error.

(2) We prepared 5 replicated libraries from one healthy PBMC gDNA sample to analyze specificity of QBDA AML panel (Section “QBDA AML panel for MRD detection” in manuscript). If a mutation is observed in ≥ 4 out of the 5 libraries, we believe this is a true positive mutation existing in the DNA sample, not an artifact caused by polymerase misincorporation or sequencing error, because the probability of the same error appearing 4 times out of 5 experiments is extremely low. After filtering out the true positives, we observed only 1 false positive mutation call out of the 5 libraries. The specificity study with technical replicate experiments further reduced the likelihood of random error.

The mutation might be originated from contamination, especially when synthetic templates were added as spike-in positive controls as the case in pan-cancer panel. Because the low frequency mutation in healthy sample due to clonal hematopoiesis, DNA damage, or contamination are all below LoD, the impact on MRD detection should be not significant.

Reviewers' Comments:

Reviewer #1:

Remarks to the Author:

I am satisfied with the edits and responses to my comments from my review. They did not address all the comments (particularly about testing FFPE samples of different quality/age for this paper but did mention they will pursue this in a future study). Stephen Yip

Reviewer #2:

Remarks to the Author:

I am happy with the revisions and appreciate the detail that the authors took in their reponse.

Reviewer #1 (Remarks to the Author):

I am satisfied with the edits and responses to my comments from my review. They did not address all the comments (particularly about testing FFPE samples of different quality/age for this paper but did mention they will pursue this in a future study). Stephen Yip

We thank the reviewer for all the comments.

Reviewer #2 (Remarks to the Author):

I am happy with the revisions and appreciate the detail that the authors took in their response.

We thank the reviewer for all the comments.